# Hypothalamic Inflammation as a Potential Pathophysiologic Basis for the Heterogeneity of Clinical, Hormonal, and Metabolic Presentation in PCOS

**DOI:** 10.3390/nu13020520

**Published:** 2021-02-05

**Authors:** Danai Barlampa, Maria Sotiria Bompoula, Alexandra Bargiota, Sophia Kalantaridou, George Mastorakos, Georgios Valsamakis

**Affiliations:** 1Unit of Endocrinology, Aretaieion University Hospital, Medical School of Athens, Ethnikon and Kapodistriakon University of Athens, Athens, Vasilisis Sofia Avenue 76, 115 28 Athens, Greece; danai_mp@hotmail.gr (D.B.); gedvalsamakis@yahoo.com (G.V.); 2Reproductive Endocrinology Unit, 3nd University Department of Obs & Gynae, Attikon University Hospital, Medical School of Athens, Ethnikon and Kapodistriakon University of Athens, 12462 Athens, Greece; bompoula@gmail.com (M.S.B.); sophiakalantaridou@gmail.com (S.K.); 3Department of Endocrinology and Metabolic Disorders, University Hospital of Larissa, Medical School of Larissa, University of Thessaly, 41334 Larissa, Greece; abargio@yahoo.gr

**Keywords:** high-fat diet, nutrients overconsumption, polycystic ovary syndrome, PCOS, hypothalamic inflammation, hypothalamus

## Abstract

Polycystic ovary syndrome (PCOS) is the most common endocrine disorder among women of reproductive age. It is a heterogeneous condition characterized by reproductive, endocrine, metabolic, and psychiatric abnormalities. More than one pathogenic mechanism is involved in its development. On the other hand, the hypothalamus plays a crucial role in many important functions of the body, including weight balance, food intake, and reproduction. A high-fat diet with a large amount of long-chain saturated fatty acids can induce inflammation in the hypothalamus. Hypothalamic neurons can sense extracellular glucose concentrations and participate, with a feedback mechanism, in the regulation of whole-body glucose homeostasis. When consumed nutrients are rich in fat and sugar, and these regulatory mechanisms can trigger inflammatory pathways resulting in hypothalamic inflammation. The latter has been correlated with metabolic diseases, obesity, and depression. In this review, we explore whether the pattern and the expansion of hypothalamic inflammation, as a result of a high-fat and -sugar diet, may contribute to the heterogeneity of the clinical, hormonal, and metabolic presentation in PCOS via pathophysiologic mechanisms affecting specific areas of the hypothalamus. These mechanisms could be potential targets for the development of effective therapies for the treatment of PCOS.

## 1. Introduction

The hypothalamus, being part of diencephalon, is situated beneath the thalamus and above the midbrain. It consists of many nuclei, with different functional roles, organized in four regions: the preoptic, anterior (supraoptic), median (tuberal), and posterior (mammillary) hypothalamus, each of which has a lateral, medial, and periventricular zone [1] (Figure 1A). The hypothalamus regulates genetically preprogrammed behaviors, such as temperature, feeding, defensive and aggressive behaviors, as well as emotional responses to stress and panic-related behavior, libido, and reproduction [2]. Inflammatory pathways can be activated in the hypothalamus. The preoptic area is known to control thermoregulation, reproduction, and electrolyte balance. In the median region, dorsomedial, ventromedial, and arcuate (infundibular) nuclei are involved in regulation of body-weight balance, food intake, satiety, thirst, and circadian rhythms. Dysfunction of these nuclei causes hyperphagia and obesity [2]. Gonadotropin-releasing hormone (GnRH) neurons are a heterogeneous population of hypothalamic neurons, which control reproduction. The majority of GnRH neuronal cell bodies are located in the arcuate nucleus (part of the mediobasal hypothalamus) and in the medial preoptic area. GnRH is secreted in a pulsatile fashion and, through the portal circulation, stimulates synthesis and secretion of luteinizing hormone (LH) and follicle stimulating hormone (FSH) from the anterior pituitary gland. These gonadotropins are responsible for the secretion of gonadal steroids, which exert a negative feedback in the brain [3].

Hypothalamic inflammation bears characteristics of chronic low-grade inflammation, notably at the molecular level [4]. On the one hand, a high-fat diet (HFD) has been associated to hypothalamic inflammation [5]. Over-nutrition leads rapidly to an inflammatory response in the hypothalamus and to a disrupted appetite [6]. In rodents, only 24–72 h following initiation of HFD, an elevation of markers of hypothalamic inflammation is observed [7]. It has been noticed that hypothalamic inflammation occurs earlier than excess weight gain and obesity do. Eventually, hypothalamic inflammation can be both the cause and the outcome of a diet-induced metabolic disease. In addition, known metabolic disorders are associated with inflammation in the peripheral organs [4]. On the other hand, diet-induced obesity induces a chronic low-grade inflammatory response in the body and is one of the most representative metabolic disorders. Over-nutrition with a high-caloric diet results in hyperplasia and hypertrophy of adipocytes. Thus, these cells lead to adipose tissue dysfunction and ectopic fat accumulation [8]. Ectopic fat storage in the liver, pancreas, skeletal muscle, and visceral adipose tissue contributes to the development of insulin resistance and inflammation in these target organs [9]. Inflammation in the pancreatic islets seen in type 2 diabetes mellitus (T2DM) includes pancreatic infiltration by immune cells, such as macrophages [10]. The excess of unsaturated fatty acids (FAs), such as palmitic acid, stimulates pancreatic islet inflammation as well [11]. Then, β-cell dysfunction and insulin resistance ensue.

Polycystic ovary syndrome (PCOS) is the most common endocrine disorder among reproductive-aged women. Its prevalence ranges from 6–20% depending on the used diagnostic criteria [12]. The Rotterdam diagnostic criteria for PCOS are the most frequently used. According to them, the PCOS definition should include two of the following three criteria: 1. clinical or/and biochemical hyperandrogenism; 2. chronic anovulation; and 3. polycystic ovarian morphology, after the exclusion of other secondary causes [13]. PCOS is a complex condition characterized not only by reproductive but also metabolic, endocrine, and psychiatric features. Patients with PCOS present reproductive (such as subfertility and infertility) and metabolic abnormalities (such as marked insulin resistance combined with elevated risk for T2DM, cardiovascular disease, hepatic steatosis, dyslipidemia, and obesity) [14]. Depression and anxiety complete the profile of this heterogeneous condition. Studies have shown that these metabolic disturbances are associated with inflammation in peripheral target organs [15]. More than one pathophysiologic mechanism is involved in the development of PCOS, including environmental, genetic, and epigenetic mechanisms. A defect in the GnRH/gonadotropins neuro-endocrine axis is also one of the potential causative mechanisms of PCOS [16]. The generation of GnRH pulses is driven by a dynamic balance between excitatory and inhibitory signals. The kisspeptinergic system is one of them and serves as regulator of GnRH pulsatility [17]. The energy status of the human body (either positive or negative) affects the kisspeptinergic system. Therefore, the metabolic elements from the periphery interfere actively with the centers of reproduction in the brain [17]. In PCOS women, insulin resistance and the concomitant hyperinsulinemia result in hyperandrogenemia thus participating in the pathogenic mechanism of PCOS [18].

The aim of this critical review is to explore whether the development of diet-induced hypothalamic inflammation could be involved with the pathophysiology in PCOS etiology and the heterogeneity of its clinical, hormonal, and metabolic presentation.

## 2. Methodology

An extensive search in electronic search engines, such as MEDLINE, PubMed platform, SCOPUS, Directory of Open Access Journals, and BioMed Central, was conducted by employing key words: hypothalamic inflammation and hypogonadism, eating disorder, amenorrhea, depression, anxiety, oligo-anovulation, anovulation, infertility, obesity, diabetes, prediabetes, glucose intolerance, insulin resistance, metabolic syndrome, metabolic disease, dyslipidemia, hypercholesteremia, lipid, and glucose. A total of 4160 publications and scientific abstracts were identified from the initial search between 2008 and 2020. After having introduced the third component (i.e., PCOS), results were narrowed down to 65. Among them, 10 were reviews, 1 was a case-control study, 1 was a cross-sectional study, 2 were cohort studies, 1 was a randomized control trial, and 1 was an animal study.

## 3. Potential Mechanisms Triggering Inflammation in the Hypothalamus

### 3.1. Glucose

In normal feeding conditions, Agouti-related protein (AgRP) and pro-opiomelanocortin (POMC) neurons in the arcuate nucleus can sense the extracellular glucose concentrations and participate, with a feedback mechanism, in glucose homeostasis [19]. An excess consumption of nutrients can trigger inflammatory pathways [20]. More specifically, a combination of fat and sugar overconsumption, but not fat overconsumption per se, leads to excessive production of major glycation end-products in hypothalamic neurons, which consequently stimulate hypothalamic inflammatory responses, such as microgliosis. This process results in neuronal dysfunction and energy imbalance [21]. In rodents, it has been shown that this inflammatory process in the hypothalamus starts hours after glucose infusion in the brain, much earlier than any visible weight gain is observed [7]. In neonatal male rats, a diet enriched in sucrose induced hypothalamic inflammation in a different way than HFD does, i.e., without the classical astrogliosis but via production of several proinflammatory factors [22].

### 3.2. Saturated Fatty Acids

A HFD containing large amounts of long-chain saturated FAs (SFAs) can induce hypothalamic inflammation and impair insulin and leptin signaling via activation of c-Jun N-terminal kinase (JNK) and nuclear factor κB (NF-κB) signaling pathways. In addition, it induces the production of known proinflammatory cytokines, such as tumor necrosis factor (TNF), interleukin (IL)-6, and IL-1β [5,23]. More specifically, palmitate and stearic acids cross the blood brain barrier and inhibit the activation of insulin receptor substrate 1 (IRS1), a known phosphoinositide 3-kinase (PI3K) signaling pathway, thus promoting insulin resistance and adiposity [24]. In contrast, when saturated FAs in HFD are replaced with unsaturated FAs, hypothalamic inflammation seems to be partially reversible while insulin and leptin resistance seem to be restored [25]. This phenomenon can be partly explained by the fact that long-chain SFAs are less efficiently oxidized than unsaturated FAs. In addition, during HFD, the imbalance between FAs uptake and utilization leads to accumulation of toxic lipid species, a phenomenon known as *lipotoxicity*, which results in increased cellular oxidation (due to overproduction of reactive oxygen species, ROS) and insulin resistance. The increased levels of ROS alter several molecular pathways, leading to neuronal damage and to hypothalamic inflammation [23]. It is extremely intriguing that in mice models, fat accumulation in the hypothalamus may conquer into sexual dimorphism [26]. This HFD-induced dysfunction is amplified by the subsequent dysregulation of the endoplasmic reticulum, which is responsible for proper protein folding. Thus, an unfolded protein response ensues, leading to insulin and leptin resistance and subsequently, to obesity and its metabolic consequences [27]. Finally, in mice, toll-like receptors (TLRs) are activated by FAs and the gut microbiome as well, participating in this way in metabolic regulation [28].

### 3.3. Uric Acid

Hyperuricemia constitutes one of the major health problems worldwide. It is estimated that 5–10% of adult Americans suffer from the complications of hyperuricemia. Elevated levels of uric acid are combined with metabolic syndrome, glucose intolerance, dyslipidemia, and obesity. In rodents, a high-uric acid diet activates the Nuclear Factor-kappa B (NF-κB) pathway, leading to the expression of proinflammatory cytokines, and increased reactive gliosis in the hypothalamus. Intracerebroventricular injection of uric acid induced inflammation and reactive gliosis in the mediobasal hypothalamus, which was confirmed by magnetic resonance imaging (MRI) both in rodents and humans [29].

### 3.4. Stress

In stress-related hypothalamic inflammation, glucocorticoids play a primordial role [30]. The hypothalamus contains many glucocorticoid and mineralocorticoid receptors, which are involved in the regulation of the hypothalamic–pituitary–adrenal (HPA) axis activity. Activation of the HPA axis results in systemic elevations of glucocorticoids, which act in order to maintain homeostasis when a stressor threatens homeostasis [31]. Chronic exposure to stress results in glucocorticoid resistance, specifically in the corticotropin releasing hormone (CRH) neurons of the paraventricular nuclei (PVN) in the hypothalamus. In turn, the negative feedback of the HPA axis is disrupted. As a consequence, a chronically activated HPA axis in combination with high serum glucocorticoids alters the physiologic response of tissues to glucocorticoids [32,33,34]. Specifically, in the hypothalamus, elevated concentrations of glucocorticoids induce the expression of ionized calcium-binding adaptor protein 1 (Iba1) (a microglial and macrophage-specific calcium-binding protein involved with membrane ruffling and phagocytosis in activated microglia), a marker of microglia activity [35]. Activated microglia are responsible for the production of several inflammatory factors [36] while it increases the cell-surface receptors, such as TLRs, which increase its susceptibility to other stimuli, i.e., lipopolysaccharides [37]. In addition, glucocorticoids might increase the maturation and release of IL-1β in response to extracellular ATP in human macrophages. IL-1β in the hypothalamus induces CRH secretion from PVN and activates the HPA axis. Moreover, stress itself might induce IL-1β expression in the hypothalamus with the above-mentioned consequences. Studies in rodents have shown that exposure to psychosocial stress increases the expression of various inflammatory markers, including IL-1β [38]. Along with this, IL-6 is a consistently elevated biomarker of chronic stress in human and animal models. Administration of recombinant IL-6 in humans has a prolonged stimulatory effect on the HPA axis, increasing circulating adrenocorticotropic hormone (ACTH) and cortisol concentrations [39]. Apparently, in chronic stress, a vicious cycle of glucocorticoid resistance, central IL-1β expression, and IL-6 and HPA axis hyperactivity is installed [40].

## 4. Heterogeneity of Clinical, Hormonal, and Metabolic Presentation in PCOS Physiopathologic Mechanisms

### 4.1. Heterogeneity of Clinical, Hormonal, and Metabolic Presentation in PCOS

PCOS is a highly prevalent endocrine disorder affecting a woman’s overall health long term. It is a complex condition with a broad spectrum of clinical manifestations and associated morbidities [41]. Women with PCOS have an increased risk for developing cardiovascular disease, metabolic syndrome, reproductive abnormalities, depression, and certain forms of cancer. A combination of clinical symptoms, such as menstrual abnormalities, acne, hirsutism, and alopecia, is reported in PCOS [42]. In a systematic review and meta-analysis, the authors reported that women with PCOS present with abnormal eating disorders three times more frequently as compared to women without PCOS [43]. The prevalence of bulimia nervosa and binge eating disorder were increased in these women. Of note, 50% of individuals with a lifetime diagnosis of bulimia nervosa and 32% with binge eating disorder have also experienced major depression, while almost 12% of those with bulimia nervosa or binge eating disorder present with generalized anxiety disorder. Indeed, 27–50% of women with PCOS report depressive symptoms compared to 19% of women without PCOS. In another study, in PCOS women, the overall prevalence of psychiatric morbidity was 50%, while specifically that of anxiety disorder was 38.6% [42].

Women with PCOS present higher androgen and estradiol (E2) concentrations, hyperinsulinemia, and suppressed sex hormone binding globulin (SHBG) concentrations compared with women without PCOS, while women with PCOS and obesity more often present ovarian dysfunction and amenorrhea [44].

Obesity is a common symptom (50% of PCOS women are overweight). By 40 years of age, up to 40% of women with PCOS will develop impaired glucose tolerance while 8–10% of PCOS patients are diagnosed with T2DM [14,45]. Women with PCOS present higher concentrations of triglycerides and low-density lipoprotein cholesterol (LDL-C) and lower levels of high-density lipoprotein cholesterol (HDL-C) than those of controls [46,47].

### 4.2. Physiopathologic Mechanisms in PCOS

A variety of physiopathologic mechanisms are involved in the pathogenesis and development of PCOS. However, the exact pathophysiological mechanism is still unknown. GnRH neurons are the neuronal denominator of a complex neuronal network, which regulates the reproductive system. A defect in the pulsatile secretion of GnRH and a subsequent elevation of LH circulating concentrations, which, in turn, enhance ovarian androgen secretion, seems to be involved in the pathophysiology of this pathologic entity [48]. Due to their anatomical position, GnRH neurons interact with a range of neuroendocrine and metabolic inputs [49]. Kisspeptin-neurokinin B and dynorphin neurons, known as KNDγ neurons, represent major regulators of GnRH pulsatility [17]. These are sited predominantly in an area corresponding to the posterior arcuate nucleus within the mediobasal hypothalamus. By classical lesion studies, this area was identified as the putative location for the GnRH pulse generator in primates [17]. GnRH neurons express the Gpr54 gene (G protein-coupled receptor GPR54), which encodes for the kisspeptin receptor, suggesting the involvement of kisspeptin in the regulation of GnRH secretion [50]. Specifically, kisspeptin and neurokinin B stimulate GnRH pulsatility while dynorphin inhibits it. Kisspeptin is considered a potent stimulator of the HPO axis [17].

Insulin resistance (predominantly in the liver, adipose tissue, and muscles) is the most prevalent metabolic perturbation in women with PCOS, affecting 65–70% of all patients [51]. It is followed by compensatory hyperinsulinemia and has been related to the reproductive defects of PCOS [18]. Hyperinsulinemia contributes to androgen-depended anovulation through distinct mechanisms. Insulin enhances ovarian androgen biosynthesis via its receptors in the theca cells [52]. The ovaries, in PCOS, do not demonstrate insulin resistance [53]. In addition, insulin increases the frequency and amplitude of GnRH pulsatility (via upregulation of GnRH gene expression in GnRH neurons) and subsequently LH pulsatile secretion. These effects contribute, in turn, to hyperandrogenemia (HA) [48]. The latter stimulates further GnRH pulse frequency by inhibiting the progesterone negative feedback, which finally leads to an increase of LH secretion [48].

The human body has to ensure a sufficient energy reservoir for reproduction. This requires a dynamic interaction between peripheral and central tissues [54]. The role of the kisspeptinergic system is to recognize the energy status of the body and to translate this information into the brain by enhancing the secretion of GnRH [17]. In detail, fasting and calorie restriction suppresses LH pulse frequency and increases LH pulse amplitude [54]. Leptin is a hormone secreted from adipose tissue and plays a crucial role in reproduction and in the regulation of LH release. In recent studies, leptin has been suggested to stimulate directly (via its own receptors) and/or indirectly kisspeptin neurons [55]. In states of low leptin activity, such as in mutations of leptin or its receptors and in functional hypothalamic amenorrhea, the release of GnRH and gonadotropins is impaired. [17,56]. The hypothalamic kisspeptin system is sensitive to changes in the body energy status and may be altered in conditions of persistent negative energy balance. However, it remains to be defined whether metabolic stress associated with an excess of energy stores, such as obesity, might have an impact on this system as well. There are controversial studies regarding the effect of diet-induced obesity on the expression of the kisspeptin mRNA (Kiss1) gene. Persistent obesity in DBA/2J mice, predisposed to reproductive disorders, is associated with marked suppression of Kiss1 mRNA in the arcuate and anteroventral periventricular nuclei [57]. However, following initiation of HFD in male Srague-Dawley rats, an increase in the expression of the Kiss1 gene in the hypothalamus has been demonstrated [58]. These findings suggest that Kiss1 neurons in both the arcuate and the anteroventral periventricular nuclei may be targets for metabolic regulation. Further research is needed regarding the exact effect of negative energy balance and metabolic stress regarding the responsiveness of the GnRH/gonadotropin system to the stimulatory effects of kisspeptin.

## 5. Hypothalamic Inflammation as a Potential Pathophysiological Mechanism of the Heterogeneity of PCOS

### 5.1. Hypothalamic Inflammation-Induced Metabolic Disorders

Several studies show that hypothalamic inflammation participates in the development and pathogenesis of metabolic diseases, such as obesity, diabetes, metabolic syndrome, hypertension, and dyslipidemia, mainly by affecting the homeostasis of food intake, energy balance, insulin and leptin signaling, and glucose metabolism in liver and fatty acid oxidation [59,60].

Key to this neuroendocrine regulation of energy balance is the melanocortin system. In particular, in a healthy state, the arcuate nucleus contains the orexigenic AgRP/neuropeptide Y (NPY) and the anorexigenic POMC/cocaine- and amphetamine- regulated transcript (CART) neurons [61,62]. Leptin and insulin, which are secreted by the adipose tissue and pancreas, respectively, cross the blood brain barrier and bind to their receptors in the arcuate nucleus. The blood vessels of this circumventricular organ are fenestrated and lack tight junctions, allowing exposure to solutes from the circulation [63]. In a fed state, leptin binds to the leptin receptor ObR/LrpR and increases POMC gene expression, thus releasing the active form of α-melanocyte-stimulating hormone (α-MSH) [64]. Then, α-MSH binds to the melanocortin 3/4 (MC3/4R) receptor in the PVN but also in the dorsomedial hypothalamic nucleus, in the lateral hypothalamus, as well as in the ventromedial hypothalamus, exerting its anorectic effects [65,66]. In contrast, in fasting state, insulin activates the AgRP promoter and represses the POMC promoter, resulting in an increase in food intake [67]. AgRP is an inverse agonist of the MC3/4R receptor and counters the anorectic effects of α-MSH [68].

As already mentioned, studies in mice showed that hypothalamic inflammation presents within few hours after initiation of HFD and especially prior to peripheral inflammation and notable weigh gain [5,7]. Since the first days, a local inflammation is noticed, in particular an activation of microglia mainly in the mediobasal hypothalamus in the arcuate nucleus, the anterior part of PVN, and in the median eminence [65] (Figure 1B). A study in mice showed the need for an adequate presence of fats and carbohydrates in the diet in order for microglia to expand. Feeding mice with a very high-fat very low-carbohydrate diet did not affect microglia cell proliferation [69]. The latter disrupts neuronal regulation of the energy balance by inducing inflammation, thus resulting in degradation of POMC neurons [70]. This inflammation promotes stress of the endoplasmic reticulum, leading to insulin and leptin resistance in the central nervous system (CNS) [65]. The increased number of activated microglia possibly contributes further to neurodegeneration [69].

Τhe destruction mainly of the PVN and ventromedial hypothalamic nuclei following inflammation leads to overeating, increased weight gain, obesity, and T2DM. The activation of several inflammatory pathways is known to induce this local inflammation. The activation of the proinflammatory JNK1 pathway in AgRP neurons increases spontaneous firing in these cells and, along with central leptin resistance, it leads to hyperphagia, weight gain, and adiposity [71]. In addition, activation of the inhibitory inflammatory IKK2 pathway mitigates the AgRP response to insulin and impairs glucose homeostasis. In another study, overactivation of the STAT/SOCS3 intracellular pathway by increased concentrations of leptin, resulting from growing adipose tissue, not only leads to leptin but also to insulin resistance at the level of CNS [72]. Further to this, mitochondria are also essential for energy regeneration and sensing of energy cellular demands via production of ROS [73]. Following HFD, impairment of the balance between production and disposal of ROS in the hypothalamus can result in endoplasmic reticulum stress, leptin resistance, and changes in eating behavior and in glucose utilization. All this inevitably leads to obesity and T2DM [70]. Similarly, production of transforming growth factor β (TGFβ) by hypothalamic astrocytes following HFD induces RNA stress, especially in POMC neurons, which affects proper function of proteins, thus leading to dysfunction of POMC [74]. An important outcome of this process is that the hypothalamus can no longer exert appropriate control on hepatic gluconeogenesis [75]. As a result, an early defect in a mechanism tightly connected to hypothalamic neuronal protein function can contribute to glucose intolerance via alteration of the hepatic physiology [75,76]. Furthermore, when autophagy in AgRP neurons is inhibited, mice eat less and are lean, whereas when autophagy is inhibited in POMC neurons, mice eat more and are obese [77,78,79]. Following HFD, autophagy activity in the mediobasal hypothalamus is impaired severely. The consequent hypothalamic inflammation further inhibits hypothalamic autophagy in lean mice, thus inducing increased caloric intake and obesity [80,81] (Figure 2).

Another potential mechanism for the development of peripheral insulin resistance and T2DM is through alteration of the activity of *autonomic nerves*, which innervate the liver, muscles, adipose tissue, and pancreas [83]. The PVN nuclei exert their regulatory influence via several mechanisms, including modulation of the activity of the sympathetic nervous system. In addition, the reduced hypothalamic insulin and glucose sensitivity along with the subsequently altered neuroendocrine regulatory mechanisms (including HPA axis activation) may also participate in the development of insulin resistance [84].

Insulin also controls fatty acids (FA) release from white adipose tissue (WAT) through direct effects on adipocytes and indirectly through hypothalamic signaling by reducing sympathetic nervous system outflow to WAT. Uncontrolled FA release from WAT promotes *lipotoxicity*, which is characterized by inflammation and insulin resistance. In support, after the initiation of a three-day HFD to Sprague-Dawley rats, a 37% increase in caloric intake and elevated base-line free FAs and insulin levels compared with controls rats were observed. The results were that overfeeding did not impair insulin signaling in WAT, but it abolished the ability of mediobasal hypothalamus insulin to suppress WAT lipolysis and hepatic glucose production. Insulin levels in Sprague-Dawley rats were moderately raised during the fed period, consistent with systemic insulin resistance [85]. Dyslipidemia could also follow the same pathophysiological pathway. However, there are still no human studies to support direct hypothalamic inflammation-induced dyslipidemia in PCOS.

### 5.2. Potential Hypothalamic Inflammation-Associated Mechanisms in Reproductive Disorders

Women with PCOS and obesity tend to have more notable endocrine disturbances [44]. They present more severe cycle disturbances and they need increased dosages of ovulation induction drugs compared to lean PCOS patients, while they demonstrate increased risk of non-response to such treatment [86]. It has been shown by several studies that acute inflammation affects reproduction. Administration of cytokines or lipopolysaccharide (LPS) intracerebroventricularly in female rats represses GnRH and LH gene expression and reduces GnRH neuropeptide release and gonadotropin concentrations [87]. In any case, obesity causes low-grade inflammation in the organism, but its effects on reproduction via GnRH neurons are to be investigated. Several studies have shown that HFD negatively affects the estrous cycle as it is associated with a high incidence of anovulation. However, it is still unknown if this is what happens in PCOS [88]. Studies in animals have shown that a high-fat and high-sugar (HFHS) diet resulted in irregularity of the estrous cycle and anovulation. Regarding follicular development, animals on the HFHS diet had significantly elevated ovarian cyst counts compared to controls [89]. In another study, advanced glycation end-product (AGEs) levels were found to be increased in the serum of young women with PCOS [90]. In addition, increased immunostaining of AGEs and their receptors (RAGE) was observed in the different compartments of the ovarian tissue in polycystic ovaries [91]. Specifically, serum concentrations of some AGEs and intraovarian deposition of AGEs could potentially contribute to alteration in oocyte function, fertilization, and embryo development [92]. Another commonly cited factor, seen in unexplained reproductive abnormalities, is chronic stress. As already mentioned, chronic stress could induce hypothalamic inflammation in mice [35]. Even in pregnancy, chronic stress can lead to elevated concentrations of cortisol and insulin [93]. It has been suspected that chronic stress, via hypothalamic inflammation, could potentially activate the hypothalamus–pituitary axis and result in reproductive, metabolic, and endocrine dysregulation.

In this review, data are presented showing that HFD can induce hypothalamic inflammation. In addition, many studies have shown that an HFHS diet is associated with menstrual disorders, anovulation, and difficulties in fertilization in women diagnosed with PCOS. Thus, it appears that diet-induced hypothalamic inflammation may contribute to the endocrine pathogenesis of PCOS. Of note, the extent and pattern of hypothalamic inflammation in these patients is not yet fully studied. The preoptic area as well as the arcuate nucleus, which modulate secretion of GnRH, could be a potential target of hypothalamic inflammation.

### 5.3. Potential Mechanisms of Hypothalamic Inflammation-Associated Psychiatric Disorders

As stated before, PCOS is associated with depression and anxiety disorders [81]. Chronic exposure to intense stress is associated with an elevation of inflammatory molecules in the hypothalamus, altering the activity of the HPA axis, ultimately leading to glucocorticoid resistance [32,33]. The latter might participate in the pathogenesis of psychiatric disorders, including depression [30,94]. Long chronic stress induces glucocorticoid resistance in CRH neurons in PVN, thus disrupting the negative feedback and altering the physiological function of the HPA axis [30]. Along with this, repeated administration of recombined IL-6 in humans leads to a blunted ACTH response [39]. This resetting of the HPA axis has also been observed in melancholic depression. Glucocorticoid resistance decreases the availability of neuroprotective factors and increases the expression of inflammatory molecules in the hypothalamus [33]. Moreover, increased glucocorticoid concentrations induce neurotoxicity and subsequent atrophy in the hippocampus, which is crucially involved in the pathogenesis of depressive disorders [95]. Overall, these changes might lead to alterations in neurotransmission, neurodegeneration, nerve cell death, and ultimately depression [96]. Indeed, recent studies indicate that hypothalamic inflammation might also be associated with stress exposure and psychiatric diseases, including depressive disorder [97].

It seems that hypothalamic inflammation could be associated with the psychiatric profile described in women with PCOS. The exact pathophysiology of hypothalamic inflammation-induced psychiatric disorders in PCOS women needs to be further elucidated.

## 6. Discussion

The anatomy of the hypothalamus is well defined. Specific regions of nuclei of the hypothalamus are associated with the control of physiological activities of the body. There is an extensive connectivity among the hypothalamic regions and nuclei as well as the hypothalamus with other brain regions. As inflammation in specific hypothalamic nuclei affects specific metabolic and reproductive centers, it is of interest to what extent these nuclei are involved in the local inflammatory expansion translated accordingly to a variety of the body’s metabolic and reproductive symptoms. Although the extent and the pattern of hypothalamic inflammation are not fully studied, its chronic presence might disrupt hypothalamic neuroendocrine pathways of the arcuate nucleus (kisspeptinergic system) and PVN, thus affecting, simultaneously, GnRH secretion and appetite [7].

An increase in the expression of proinflammatory mediators, particularly in the medio basal hypothalamus, is observed 24 h following HFD administration to rats. Specifically, in rats, TNF, IL-1β, and IL-6 levels, following an initial 3-day increase, normalize, only to rise again in 2–3 weeks and remain elevated thereafter. It is of interest that this pattern of increase of these proinflammatory mediators parallels that of the administrated diet. In addition, neurons in the hypothalamus initiate a stress response within the first week of HFD administration, close to the onset of hypothalamic inflammation. The accumulation of reactive astrocytes and microglia in the mediobasal hypothalamus occurs around the same time and partly reduces inflammation. In addition, in the same study, it has been shown that 8 months of HFD administration resulted in a 25% decrease in POMC neurons, via increased autophagy. Thus, it is suspected that chronic exposure to HFD can cause permanent damage to neurons while it is not known to what degree the neuronal population can react [7].

Furthermore, studies in mice reveal that HFD-induced obesity is associated with an increased number of Iba-1-positive cells [98]. Iba-1 is a protein that is essentially expressed in all microglia in the brain [99]. It is widely used as an immunohistochemical marker for both ramified and activated microglia. The latter is detected particularly around the *organum vasculum* of the *lamina terminalis* (OVLT) in the rostral hypothalamus and in the median eminence and the arcuate nucleus but neither in the lateral preoptic area nor in the PVN. Thus, one could assume that the expression of hypothalamic inflammation is not uniform and that it has a variable pattern [98].

It is still unknown whether the extent of this inflammation to adjacent tissues in the brain could be a reliable predictive factor for the development of metabolic and endocrine diseases in PCOS. It is well established that an accurate integration of metabolic stimuli by the hypothalamic–pituitary–gonadal axis is crucial for a normal reproductive state [56]. GnRH neurons integrate all signals in the brain to regulate reproduction [100]. A multitude of factors (such as RFamide-related peptide 3 (RFRP-3), a gonadotropin-inhibitory hormone, senktide, a neurokinin B receptor agonist, and oxytocin), involved in reproductive function, act within the hypothalamus as effectors upon *Gnrh* mRNA transcription and consequently GnRH secretion. Of note, GnRH neurons express cytokine receptors. Thus, *Gnrh mRNA* may be repressed via activation of cytokine receptor signaling pathways [98]. GnRH neurons are specifically located close to OVLT. The preoptic area that surrounds the OVLT contains fenestrated capillaries [63,101,102] and the permeability of the brain blood barrier in this specific region is greater, making it more vulnerable to proinflammatory cytokines, immune cell infiltration, and activated microglia [98].

Given this, location constitutes one of the major determinants of the neuro-immune alteration of the GnRH neurons and potentially could illuminate pathogenetic mechanisms in PCOS. Of note, ovarian steroidogenesis is directly related to the hypothalamus–pituitary axis and alterations of LH secretion are involved directly in the physiopathology of PCOS. Indeed, in PCOS women, an insensitivity of the hypothalamic GnRH physiology to progesterone leads to excess LH secretion and then to the impairment of follicle maturation [103].

## 7. Conclusions

In this review, we explored whether hypothalamic inflammation could represent a common pathophysiologic basis for the heterogeneous clinical, hormonal, and metabolic presentation of PCOS.

Thus, an HFHS diet induces low-grade inflammation in the hypothalamus and more specifically in the arcuate nucleus within the mediobasal hypothalamus, in the anterior part of PVN, and in the median eminence [5]. The presence of fats and carbohydrates in the diet leads to microglia expansion and to hypothalamic inflammation [69]. Hypothalamic inflammation results in over-eating, weight gain, insulin resistance, and consequently T2D and obesity.

As females diagnosed with PCOS are at increased risk of developing cardiovascular and metabolic diseases, reproductive disorders, and depression, it could be suggested that HFD and the resulting hypothalamic inflammation could be part of the pathophysiologic origin of the syndrome. Women with PCOS and obesity tend to have more notable endocrine disturbances, which result in ovarian dysfunction and a greater rate of oligo/amenorrhea compared to lean women with PCOS. Furthermore, HFD negatively affects the menstrual cycle, with a high incidence of anovulation. It is still unknown whether HFD affects anovulation in PCOS [44,88,104].

In addition, hypothalamic inflammation might be induced by exposure to chronic stress and, in turn, might participate in the pathogenesis of psychiatric diseases, including depressive disorder. Specifically, increased concentrations of glucocorticoids, as encountered in intense stress, may induce neurotoxicity and subsequent atrophy of the hippocampus, a brain area crucially involved in the pathogenesis of depressive disorders [95].

The well-known anatomy of the hypothalamus implies increased connectivity between the hypothalamic regions and nuclei in proximity. As already mentioned, GnRH neurons are the integrators of brain stimuli regulating reproduction. They are located close to OVLT, which contains fenestrated capillaries. The permeability of the blood brain barrier in this specific region is greater than in other areas of the brain, thus resulting in increased vulnerability to proinflammatory cytokines, immune cell infiltration, and activated microglia, which ultimately can affect negatively the reproductive system [98,101,102]. It is possible that the preoptic area could be simultaneously affected by inflammation in the neighboring POMC and NPY metabolic neurons, which are located in the arcuate nucleus. However, this depends mainly on the extension of the inflammation in the brain area. It is still unknown whether the extension of this inflammation to adjacent target regions constitutes a possible factor for the development of metabolic and endocrine diseases, insulin resistance, glucose intolerance, T2DM, and obesity. According to the extent of the inflammation and the regions affected, PCOS could present with a variety of clinical, hormonal, and metabolic profiles. The factors that determine the extent of inflammation in this area of the hypothalamus are still unknown. Potential cofactors could be anatomical boundaries or other, still unknown, molecular pathways.

Treatment attenuating hypothalamic inflammation might be useful. A diet rich in unsaturated fatty acids (UFAs), such as omega −3 and −9, instead of SFAs can reverse inflammation in the hypothalamus. An UFA diet is anti-inflammatory by increasing anti-inflammatory proteins and by improving leptin and insulin signaling [25]. In the hypothalamus of mice receiving HFD, the activation of polyunsaturated fatty acid receptors (GPR120) and free fatty acid receptor 1 (GPR40) may also participate in the reduction of hypothalamic inflammation via a decrease of the proinflammatory cytokines IL-1β and TNF and an increase of the anti-inflammatory cytokine IL-10 [105]. Caloric restriction may also represent another efficient approach [106]. Several studies also support the role of diets rich in polyphenols, to modulate the inflammatory response and oxidative stress in the hypothalamus [107]. Moreover, exercise has a beneficial effect in the reduction of body weight alone but also in tempering the installed hypothalamic inflammation by decreasing microglia activation and by improving glucose intolerance [108].

When lifestyle modification is not sufficient, hypothalamic inflammation may also be attenuated by drugs. One promising therapeutic agent could be glucagon-like peptide-1 (GLP-1) analogs. The latter, currently used in the treatment of T2DM, improve glycemic control and insulin resistance together with weight loss while they have neuroprotective and anti-inflammatory effects [109]. Given the importance of these effects along with the actions of GLP-1 analogs in correcting hyperandrogenemia and oligo/amenorrhea, these pharmaceutical agents could be key in PCOS treatment [110].

Further research is recommended in humans using the methodology of brain imaging, such as functional MRI and spectroscopy, to study the extent and the pathophysiologic effects of hypothalamic inflammation in the heterogeneity of the clinical presentation of PCOS. A more specific definition of hypothalamic inflammation and implementation of imaging criteria of brain inflammation would help greatly in research in HSHF diet-induced hypothalamic inflammation as well as research in hypothalamic inflammation-associated PCOS pathophysiology. Treatments based on lifestyle modification find their justification in this pathophysiologic approach. Further research in the type and length of diet is mandatory.

## Figures and Tables

**Figure 1 nutrients-13-00520-f001:**
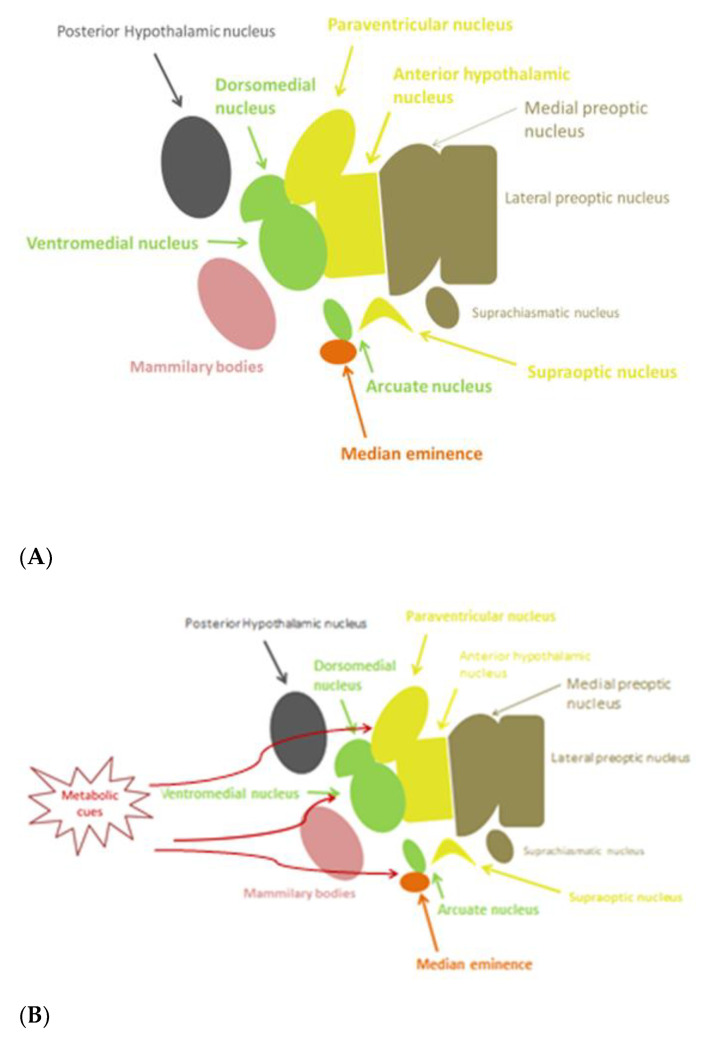
(**A**) Anatomy of hypothalamic nuclei; (**B**) Areas affected by metabolic cues in the hypothalamus.

**Figure 2 nutrients-13-00520-f002:**
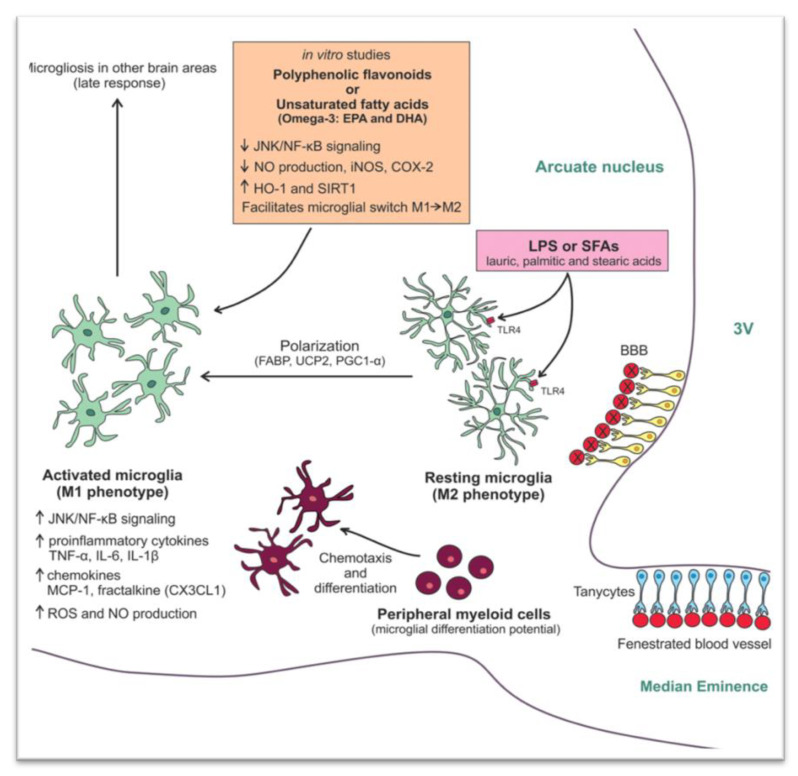
Mechanisms involved in the high fat diet (HFD) -induced hypothalamic microgliosis at molecular level [82]. Reproduced with permission from Licio A. Velloso, Hypothalamic Microglial Activation in Obesity: A Mini-Review; published by frontiers in Neuroscience, 2018.

## Data Availability

Exclude.

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
