# Peer review of "Hypothalamic Inflammation as a Potential Pathophysiologic Basis for the Heterogeneity of Clinical, Hormonal, and Metabolic Presentation in PCOS"

_nutrients, 2021, doi:10.3390/nu13020520_

Round 1

Reviewer 1 Report

The theme of this manuscript is the role of the hypothalamic inflammation in PCOS. The topic is interesting. However, several points should be addressed.

1) The abstract should be rewritten. In general, the manuscript should be checked by native English speaker.

2) Explicit the abbreviations as GnRH, PCOS, LH…

3) There are serious problems with bibliographic references. The format of the references in the text is different. Authors should check carefully the bibliography.

4) The line of reasoning of this review is difficult to follow. There is a lot of bibliographic work. But the content of the text needs to be better organized. Authors must have a clear line because the reader is completely lost.

5) I am amazed that the authors do not give informations about the kisspeptinergic system

6) Authors should illustrate the text with figures

Author Response

We thank the reviewer for the comments. We have done a lot of modifications in the manuscript, so you may find differences in the number of the pages.

Point 1: Reviewer 1 states that ‘the abstract should be rewritten. In general, the manuscript should be checked by native English speaker.’

Response 1: We thank Reviewer 1 for this comment. We have rewritten the abstract and the manuscript has been checked by native English speaker.                                                             

Point 2: Reviewer 1 states that ‘Explicit the abbreviations as GnRH, PCOS, LH….’

Response 2: We thank Reviewer 1 for this comment. We have checked the manuscript and we have explicated all the abbreviations which are in the text.

Point 3: Reviewer 1 states that ‘There are serious problems with bibliographic references. The format of the references in the text is different. Authors should check carefully the bibliography.’

Response 3: We thank Reviewer 1 for this comment. We have checked the manuscript and we have corrected the format of the references in the text.

Point 4: Reviewer 1 states that ‘The line of reasoning of this review is difficult to follow. There is a lot of bibliographic work. But the content of the text needs to be better organized. Authors must have a clear line because the reader is completely lost.’

Response 4: We thank Reviewer 1 for this comment. We understand the reviewer’s concern about the organization of the article and for this reason we have changed the article’s format.

Point 5: Reviewer 1 states that ‘I am amazed that the authors do not give information about the kisspeptinergic system’

Response 5: We thank Reviewer 1 for this comment. We have added information about the kisspeptinergic system’

Point 6: Reviewer 1 states that ‘Authors should illustrate the text with figures’

Response 6: We thank Reviewer 1 for this comment. We have added figures which illustrate the text.

Reviewer 2 Report

The submitted paper presents a review of potential implications of diet/obesity-related chronic low-grade inflammation in the hypothalamus in the context of the of polycystic ovary syndrome (PCOS). As PCOS is the most common endocrinopathy among women of reproductive age with marked heterogeneity regarding its reproductive, endocrine, metabolic, and psychological manifestations/comorbidities, the authors focus specifically on how the hypothalamus may be implicated in the pathophysiology of PCOS in relationship to high fat/sugar diet, obesity and the related chronic low-grade pro-inflammatory state.

The submitted paper has reviewed the relevant available literature and is up-to-date without major issues.

Minor points which could be addressed are the following:

1) As this review does not follow the protocol and reporting format of a systematic review, please revise in the abstract the phrase “…In this systematic review…” to something like “…In this review, …” or “…In this narrative review, …”

2) In page 1: please revise the phrase “…The Rotterdam diagnostic criteria for PCOS are the most recently used…” to something like “…The Rotterdam diagnostic criteria for PCOS are the most frequently used….”

3) In page 2: please revise the phrase “Main representative constitutes diet-induced obesity which is linked with a chronic low-grade inflammatory response…”

4) In page 2: please revise the phrase “…Hypothalamic inflammation is a chronic low-grade inflammation not bearing the classic characteristics of pain, heat, swelling or redness….”

5) In page 2: please revise the phrase “…It is astonishingly interesting that in such a small part of the brain so many important regulations of functions of the whole body take part. Coming to the subject of this review, high fat diet (HFD) has been noticed to generate low grade hypothalamic inflammation…”

6) In page 3: please revise “…systemic reviews…” 

7) In page 4: please revise the phrase “…Hyperuricemia characterizes the modern life…”

8) In page 6: please check the ref. citing format for (Shadel and Horvath, 2015) and (Han et al., 2016)

9) In pages 6 and 8: please rephrase “…Obese women with PCOS…” to “…women with PCOS and obesity…”

Author Response

We thank the reviewer for the comments. We have done a lot of modifications in the manuscript, so you may find differences in the number of the pages.

Point 1: Reviewer 2 states that ‘As this review does not follow the protocol and reporting format of a systematic review, please revise in the abstract the phrase “…In this systematic review…” to something like “…In this review, …” or “…In this narrative review, …”

Response 1: We thank Reviewer 2 for this comment. We now changed our statement that ‘In this systematic review’ and we replaced it with statement “In this review’ in abstract as per Reviewers 2 suggestion.

Point 2: Reviewer 2 states that ‘in page 1: please revise the phrase “…The Rotterdam diagnostic criteria for PCOS are the most recently used…” to something like “…The Rotterdam diagnostic criteria for PCOS are the most frequently used….”

Response 2: We thank Reviewer 2 for this comment. We now changed our statement that “…The Rotterdam diagnostic criteria for PCOS are the most recently used…” and we replaced it with statement “…The Rotterdam diagnostic criteria for PCOS are the most frequently used….” in page 2 as per Reviewers 2 suggestion.

Point 3: Reviewer 2 states that ‘in page 2: please revise the phrase “Main representative constitutes diet-induced obesity which is linked with a chronic low-grade inflammatory response…”’

Response 3: We thank Reviewer 2 for this comment. We now changed our statement that “Main representative constitutes diet-induced obesity which is linked with a chronic low-grade inflammatory response…” and we replaced it with statement “Diet-induced obesity induces chronic low-grade inflammatory response and is one of the most representative metabolic disorders ‘’ in page 2.

Point 4: Reviewer 2 states that in page 2: please revise the phrase “…Hypothalamic inflammation is a chronic low-grade inflammation not bearing the classic characteristics of pain, heat, swelling or redness….”

Response 4: We thank Reviewer 2 for this comment. We now changed our statement that “…Hypothalamic inflammation is a chronic low-grade inflammation not bearing the classic characteristics of pain, heat, swelling or redness….” and we replaced it with statement “…Hypothalamic inflammation bears characteristics of chronic low-grade inflammation, notably at molecular level ….” in page 2.

Point 5: Reviewer 2 states that ‘’in page 2: please revise the phrase “…It is astonishingly interesting that in such a small part of the brain so many important regulations of functions of the whole body take part. Coming to the subject of this review, high fat diet (HFD) has been noticed to generate low grade hypothalamic inflammation…”

Response 5: We thank Reviewer 2 for this comment. We now changed our statement that  “…It is astonishingly interesting that in such a small part of the brain so many important regulations of functions of the whole body take part. Coming to the subject of this review, high fat diet (HFD) has been noticed to generate low grade hypothalamic inflammation…” and we replaced it with statement ‘The aim of this critical review is to explore whether the development of diet-induced hypothalamic inflammation could be involved with the pathophysiology in   PCOS etiology and the heterogeneity of its clinical, hormonal and metabolic presentation.’’ in page 2 and 3

Point 6: Reviewer 2 states that in page 3: please revise “…systemic reviews…” 

Response 6: We thank Reviewer 2 for this comment. We now changed our statement that “…systemic reviews…” and we replaced it with statement “…reviews…” in page 3.

Point 7: Reviewer 2 states that ‘’ in page 4: please revise the phrase “…Hyperuricemia characterizes the modern life…”

Response 7: We thank Reviewer 2 for this comment. We now changed our statement that “…Hyperuricemia characterizes the modern life…”and we replaced it with statement “…Hyperuricemia constitutes one of the major health problems worldwide…” in page 4.

Point 8: Reviewer 2 states that ‘’ In page 6: please check the ref. citing format for (Shadel and Horvath, 2015) and (Han et al., 2016)

Response 8: We thank Reviewer 2 for this comment. Indeed there was a mistake with the references and we have corrected them.

Point 9: Reviewer 2 states that ‘’ in pages 6 and 8: please rephrase “…Obese women with PCOS…” to “…women with PCOS and obesity…”

Response 9: We thank Reviewer 2 for this comment. We now changed our statement that “…Obese women with PCOS…”and we replaced it with statement “…women with PCOS and obesity…” in page 5, 7 and 9 as per Reviewers 2 suggestion.

Round 2

Reviewer 1 Report

There is a big mistake with the pictures that are not originals. You need to have permissions because of copyright.

Author Response

We thank the reviewer for the comments

Point 1: Reviewer 2 states that ‘There is a big mistake with the pictures that are not originals. You need to have permissions because of copyright.’

Response 1: We thank Reviewer 2 for this comment. We have now created Figure 1A and B with title Figure 1. (A) ‘Anatomy of hypothalamic nuclei’ and figure 1(B) ‘Areas affected by metabolic cues in hypothalamus.’ Moreover, we have got permission for Figure 2 with title ‘Mechanisms involved in the HFD-induced hypothalamic microgliosis at molecular level.’ from Professor Licio A.Velloso. We send the relative email to Ms Selena Liu.